# Semantic Unsupervised Automatic Keyphrases Extraction by Integrating Word Embedding with Clustering Methods

**Isabella Gagliardi *** and **Maria Teresa Artese**

Institute for Applied Mathematics and Information Technologies "Enrico Magenes" (IMATI), National Research Council—CNR, Via Bassini, 15, 20133 Milan, Italy; artese@mi.imati.cnr.it

**\*** Correspondence: gagliardi@mi.imati.cnr.it; Tel.: +39-02-23699487

**Abstract:** Increasingly, the web produces massive volumes of texts, alone or associated with images, videos, photographs, together with some metadata, indispensable for their finding and retrieval. Keywords/keyphrases that characterize the semantic content of documents should be, automatically or manually, extracted, and/or associated with them. The paper presents a novel method to address the problem of the automatic unsupervised extraction of keywords/phrases from texts, expressed both in English and in Italian. The main feature of this approach is the integration of two methods that have given interesting results: word embedding models, such as Word2Vec or GloVe able to capture the semantics of words and their context, and clustering algorithms, able to identify the essence of the terms and choose the more significant one(s), to represent the contents of a text. In the paper, the datasets used are presented, together with the method implemented and the results obtained. These results will be discussed, commented, and compared with those obtained in previous experimentations, using TextRank, Rapid Automatic Keyword Extraction (RAKE), and TF-IDF.

**Keywords:** unsupervised automatic keyword extraction; clustering algorithms; word embedding models; Italian datasets; information retrieval; evaluation; word2vec; GloVe

## 1. Introduction

Increasingly, the web produces massive volumes of texts, alone or associated with images, videos, photographs, together with some metadata, indispensable for their finding and retrieval. The problem of extracting or associating keywords to texts available on the web, able to represent their contents, is a very topical problem, which requires ad hoc solutions, especially in the case of multilingual texts or specialized topics.

The keywords and key phrases of a document, if appropriately chosen, ensure the document content to be adequately and semantically represented [1,2]. Text mining, natural language processing, and information retrieval rely on keywords extraction as they provide compact information about the document and allow documents to be indexed, classified, summarized, and filtered automatically. The process of extracting keywords involves text processing methods and can be done manually by experts in the field (e.g., museum curators in the case of Cultural Heritage), or automatically by identifying and scoring words that characterize the document. The keywords/phrases obtained reflect the content and are useful to perform queries on document databases and to "suggest" similar documents.

They should, therefore, represent the content of the document to which they are associated in all its aspects and be general enough to represent several elements and, at the same time, specific enough not to represent only the whole set of items.

The problem of extracting relevant, semantic keywords has gained increasing interest over the years, also due to the enormous amount of interesting information available on the web and has been faced many times. Although the problem of the automatic extraction of keywords able to represent the content of texts has been dealt with since the early Information Retrieval systems [1,2], the advent of new tools and techniques makes it very current [3–7]. Different algorithms, categorized into supervised or unsupervised methods, have been developed to solve the problem of automatic extraction of keyphrases. Keyphrase extraction methods in unsupervised approaches can be grouped into statistical-based, graph-based, and cluster-based approaches. In statistical-based approaches, texts are usually represented as matrices in which the statistical techniques are applied to rank the words by using TF-IDF (short for term frequency–inverse document frequency) term weighting [2]. In graph-based methods, each document is represented as a graph where vertices or nodes represent words, and edges are connected based on either lexical or semantic relations, such as a co-occurrence relation. Examples are TextRank [8], Rapid Automatic Keyword Extraction (RAKE) [9], CollabRank [10], or SingleRank [11].

Derived from the PageRank algorithm, TextRank determines the importance of a word through the importance of linked words and iteratively calculates the importance of each word in the graph to select top-ranked words as keywords.

RAKE, a short form for Rapid Automatic Keyword Extraction, is an unsupervised, domain-independent, and language-independent graph-based method for extracting keywords from individual documents. The RAKE algorithm has better performance on long keyphrase extraction compared to TextRank.

Cluster-based methods extract terms, group them into clusters based on some semantic relatedness using co-occurrence similarity measures and select phrases that contain one or more cluster centroids. Examples are KeyCluster [12] or SemCluster [13].

Other good approaches to deal with Natural Language Processing (NLP) problems are Latent Semantic Analysis (LSI) [14,15] and Latent Dirichlet Allocation (LDA) [16]. The idea behind LSI is to compare words to find relevant documents, but what you really want is to compare the meanings or concepts of words. LSI is a technique able to analyze relationships between a set of documents and the terms they contain by producing a set of concepts related to the documents and terms. LDA is similar to LSI, and the process of checking topic assignment is repeated for each word in every document, cycling through the entire collection of documents multiple times. This iterative updating is the key feature of the LDA to generate a final solution with coherent topics. In both cases, documents are "clustered", and each cluster is assigned a set of keywords, to describe its meaning. The number of "clusters" has to be assigned ahead.

Table 1 compares different automatic unsupervised keyword/keyphrases extraction methods available in the literature, with the one presented in this paper, pointing out their strengths and weaknesses. One of the problems that are shared by the state-of-the-approaches is the lack of attention paid to the semantic relevance of keywords, as well as the synonyms of extracted keywords, addressed by the proposed method.

The paper presents a novel method to address the problem of the automatic unsupervised extraction of keywords/phrases from texts, expressed both in English and in Italian. The main feature of this approach is the integration of two methods that have given interesting results: word embedding models, such as Word2Vec or GloVe able to capture the semantics of words and their context, and clustering algorithms, able to identify the essence of the terms and choose the more significant one(s), to represent the contents of a text. The approach incorporates word embedding resources to improve the semantic relatedness of key phrases.

The integration of word embedding models with clustering algorithms is now being studied by different research groups, mainly with classification purposes, and/or in specific contexts (for example [17,18]) due to its ability in extracting semantic and discriminative keywords.

Our approach can be easily transferred in other contexts and languages: the use of pre-trained models allows the use even on small datasets, clustering algorithms are standard and easily adaptable. The pipeline identified for the Italian language allows an easy adaption to other languages such as French, German, etc. A phase of fine-tuning in other languages is needed to verify the results, evaluating in each step the quality of the output, and, if necessary, identifying other tools that may work better.

**Table 1.** Comparison of different unsupervised keyphrase extraction methods.

| Method | Pros | Cons |
|---|---|---|
| TF-IDF | ■ easy to compute<br>■ term-weighting improves the quality<br>■ cosine ranking formula sorts documents according to the degree of similarity to the query | ■ based on the bag-of-words (BoW) model, therefore it does not capture the position in the text, semantics, co-occurrences in different documents, etc.<br>■ assumes independence of index terms |
| TextRank | ■ based on Google PageRank<br>■ finds keywords that appear "central" (like PageRank)<br>■ can apply to any text without prior training<br>■ domain-independent | ■ extracts a few numbers of keywords, ignoring their semantic relevance<br>■ works on a single document at a time<br>■ high computational complexity. |
| Rake | ■ can apply to any text without prior training<br>■ domain-independent<br>■ works on each single document<br>■ very fast and the complexity is low.<br>■ easy to implement. | ■ needs a comprehensive and reliable list of stop words<br>■ works on a single document at a time<br>■ cannot extract semantically meaningful keywords |
| LSI | ■ documents and words are mapped to the same concept space.<br>■ concept space has vastly fewer dimensions compared to the original matrix.<br>■ the dimensions contain the most information and least noise. | ■ LSI depends heavily on SVD which is computationally intensive and hard to update as new documents appear<br>■ requires the number of clusters in input |
| LDA | ■ similar to LSI<br>■ probabilistic model | ■ requires the number of clusters in input<br>■ extracts general keywords |
| WeC<br>This method | ■ integrates two methods, which can have different variants<br>■ can apply to any text without prior training<br>■ only the preprocessing phase depends on the language of the documents<br>■ -can be applied on small datasets<br>■ easy to implement<br>■ extract semantic keywords | ■ terms are not weighted |

The paper is structured as follows: in Section 2 our approach is described in full details with a brief description of datasets used, then follows the experimentation performed on 4 different datasets in two languages, Italian and English, with some preliminary results, and a brief discussion to comment

the results, compare the different solutions and evaluate the results with baseline solutions, as Rake, TextRank, TF-IDF. Then conclusion and future works are presented.

## 2. Materials and Methods

The datasets used to test and evaluate the method were collected from open-source data, mainly from the Kaggle (https://www.kaggle.com/datasets) website and the whole web. The chosen datasets share the characteristics of having textual descriptions and, if possible, semantic tags. For the tests reported here, only the first records were used. Different clustering algorithms, word embedding models that use single or compound words, related to Italian and English languages were examined: the different options were tested and evaluated individually, as shown below.

The (multistep) method presented in this paper involves the use of clustering methods on word embedding models related to candidate terms to identify the most significant and then select those terms present as keywords or keyphrases. Starting from datasets in Italian and English, after a first pre-processing phase, which aims to eliminate or limit unnecessary or noisy information, we present the results of the unsupervised keyword extraction method. Previous experiments of the authors [19,20] have concerned classic keyword extraction algorithms: TextRank, Rake, Latent Semantic Indexing, LDA. Here we focus on cluster-based methods, such as KeyCluster [11], SemCluster [12]. Using cluster-based methods, terms are grouped in clusters based on a form of similarity or correlation: in a case like ours, where the items are words, similarity metrics are based on semantic correlation using Wikipedia and / or other measures of co-occurrence similarity. Candidate keywords containing the centroid or the terms closest (similar) to the centroid (or in the case of compound words, terms that contain the centroid) are then selected as keywords or keyphrases.

Figure 1 shows the scheme of the proposed method, whose steps (the same defined in [12]) are as follows:

S1.　Preprocessing and candidate term selection: we first preprocess the texts and select the candidate terms for keyphrase extraction, in the form of single or compound words.

S2.　Term relatedness computation: we identify a way to compute the semantic relatedness of candidate terms, based on word embedding and we use measures to produce semantic relatedness matrices.

S3.　Term clustering: based on term relatedness, we group candidate terms into clusters and find the terms (closest to the centroids) of each cluster.

S4.　Keyphrases identification: finally, we use these terms to extract keyphrases from the document.

In the following, each step is described in more depth, with some theoretical details.

### 2.1. Preprocessing and Candidate Keywords Extraction

The purpose of the preprocessing step is the identification of the set of terms, single words or compound words, which can be tagged as keywords or key phrases, in the next steps of the method.

In this pre-processing phase, several factors can influence the results:

- The language used. The experiments conducted use texts in Italian and in English.
- The grammatical form of individual or compound terms. For individual terms, words are nouns; in the case of compound terms usually a sequence of nouns and adjectives. For the English language, compound terms have the form of:

$$\text{<JJ>* <NN.*>+ <IN>)? <JJ>* <NN.*>+} \tag{1}$$

Italian allows you to construct sentences in a freer way: you can, therefore, have compounds of nouns and adjectives, with the adjective that is put before or after, without changing its meaning. Italian compound keywords will be defined by, for example:

$$\text{<ADJ>*<NOM|NPR>+<ADJ>*(<PRE><NOM|NPR>)*} \tag{2}$$

according to the Italian tagset used in the TreeTagger [21], where ADJ are adjectives, NOM and NPR are nouns, and PRE are prepositions.

- The POS tagging tool used has an important influence on how reliable the keywords extracted are.

    The preprocessing pipeline is composed of:

1.  Tokenization, i.e., the division of the text into single/multi words.
2.  Annotation, which may include part-of-speech (POS) labeling.
3.  Normalization: lemmatization/stemming, using algorithms specific for the language.
4.  Removal of stop words and/or specific grammatical categories.

While tokenization and stop word removal do not present particular problems—they simply need to be adapted to the language—finding POS tagging and lemmatization/stemming tools that work effectively with a low error rate for the Italian language is still difficult. As we said, the proposed method can be easily adapted to other languages, such as French or German: also in these languages, identification of lemmatization and POS tagging tools is the step that requires more effort to achieve good results. The tools that have been tested and the ones that have been chosen are described in detail below.

The results produced by this phase are a list of terms (even repeated), in canonical forms, such as entries in the specific vocabulary of the dataset. The set of terms can be composed of single words and compound terms, together with their POS tagging.

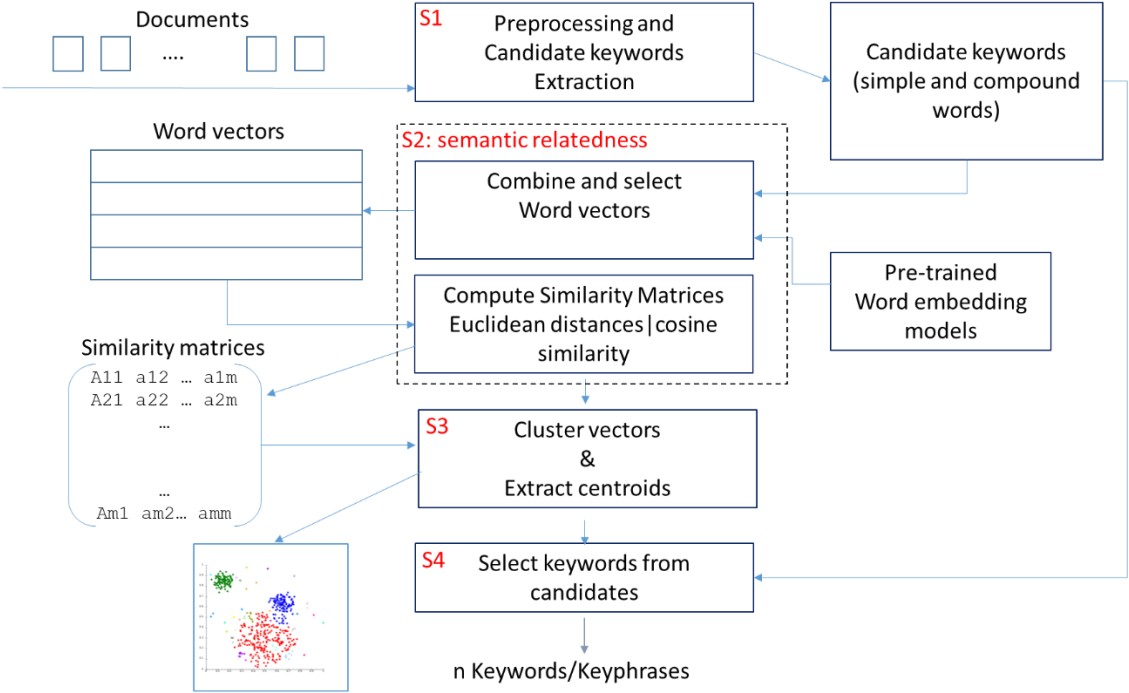

**Figure 1.** Schema of the method presented.

## 2.2. Term Relatedness Computation

The relatedness of the candidate terms has been defined and evaluated according to a measure of similarity capable of grasping their semantic content, also considering the context in which the terms are used. Different approaches for calculating the semantic similarity of terms are presented in the literature. Relatedness based on co-occurrences is an intuitive method to measure the similarity of terms present in a document within a dataset, counting co-occurrences within a window of maximum w words in the whole document, with w usually set between 2 and 20, depending on the length of

the documents. Other methods use external resources that imitate human knowledge bases, such as Wikipedia or WordNet, to measure the correlation between terms [22,23].

Here we present a method to calculate semantic relatedness based on word embedding. Word embedding technique is one of the most popular ways to represent terms, in which words with similar meanings have a similar representation, being able to capture the context of a word in a document, the semantic and syntactic similarity, the relation with other words, etc.

Each word is represented as a real value vector in a predefined vector space, and the vector values are learned in a way that resembles a neural network, so the technique is often inserted in the field of deep learning.

The key idea is to use a dense distributed representation for each word: each word is represented by a vector of tens or hundreds of dimensions, extremely reduced with respect to the thousands or millions of dimensions required for sparse word representations.

Different models have been developed since 2013 when the first models appeared: we use here word2vec and GloVe.

Word2Vec is one of the most used techniques to learn word embedding using shallow neural network [24,25].

Word2vec can utilize two model architectures to produce the word embedding model: continuous bag-of-words (CBOW) or continuous skip-gram, as shown in Figure 2. With CBOW architecture, the model predicts the current word from a window of surrounding context words. The order of context words does not influence prediction (bag-of-words assumption). In the continuous skip-gram architecture, the model uses the current word to predict the surrounding window of context words. According to the Mikolov note, CBOW is faster while skip-gram is slower but does a better job for infrequent words. The models are focused on learning about words given their local usage context, where the context is defined by a window of neighboring words. This window is a configurable parameter of the model.

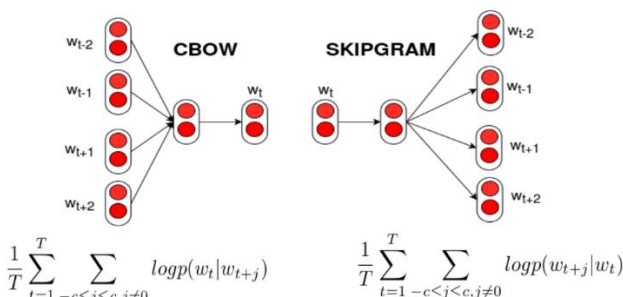

**Figure 2.** Continuous bag-of-words (CBOW) and Skipgram architecture. (https://commons.wikimedia.org/ author: Aelu013).

The Global Vectors for Word Representation, or GloVe [26], algorithm is an extension to the Word2Vec method for efficiently learning word vectors.

The result of the computation is the "translation" of the candidate terms into vectors, able to take into account the semantic content of each term, its context, the relationship with other terms, etc.

Particular attention has been paid to compound words, which are not present in the compound form in the word embedding model, but as individual components.

In this case, we consider a compound word, composed of words $w1, w2, \ldots, wn$. Each word has a word embedding vector $v_{w1}, v_{w2}, \ldots, v_{wn}$. We define the compound embedding as (https://stats.stackexchange.com/q/318885):

$$v_{cw} = \frac{1}{n} \sum_{i=1}^{n} v_{wi}. \tag{3}$$

Semantic Relatedness Matrices

The semantic relatedness of words (through their representation in the form of word embedding vectors) has been computed using standard similarity measures, such as Cosine similarity measure and Euclidean distance, to produce semantic relatedness matrices.

Formally, the Euclidean distance is also known as the Euclidean norm, L2 norm, or L2 distance and is defined as the shortest straight-line distance between two points. Mathematically this can be denoted as:

$$ed(a,b) = \sqrt{\sum_{1}^{n} (a_i - b_i)^2} \tag{4}$$

where the two points *a* and *b* are vectorized text terms in our scenario, each having length *n*.

Considering we have two terms such that they are represented in their vectorized forms, Cosine similarity gives us the measure of the cosine of the angle between them when they are represented as non-zero positive vectors in an inner product space. Thus, term vectors having similar orientation will have scores closer to 1, indicating the vectors are very close to each other in the same direction (near to zero-degree angle between them). Term vectors having a similarity score close to 0 indicate unrelated terms with a near orthogonal angle between then. Term vectors with a similarity score close to −1 indicate terms that are completely oppositely oriented to each other.

$$Cos\theta = \frac{\vec{a} \cdot \vec{b}}{\| \vec{a} \| \cdot \| \vec{b} \|} = \frac{\sum_{1}^{n} a_i b_i}{\sqrt{\sum_{i}^{n} a_i^2} \sqrt{\sum_{i}^{n} b_i^2}} \tag{5}$$

## 2.3. Clustering

Clustering is meant to assign objects in groups so that objects in the same cluster are more similar to each other than any object in any different cluster [27]. In this paper, we use three widely used clustering algorithms: Affinity Propagation, K-means, and hierarchical clustering to group the embedded vectors in clusters semantically equivalent.

Affinity Propagation (AP) is a clustering algorithm that identifies a set of 'exemplars' representing the dataset [28]. The AP algorithm is based on the concept of "message passing" among the various data points to be clustered, and no request is needed about the number of possible clusters. The input of Affinity Propagation is the pair-wise similarities between each pair of data items. In our case, the items are the vectors and the pair-wise similarity is a similarity matrix. Any kind of similarity is acceptable, thus making the Affinity Propagation widely applicable. The number of clusters is calculated by the algorithm, without the need to assign it a priori, by trial and error methods. The number of clusters identified by the Affinity Propagation algorithm is also used by the other algorithms used in the experiments in the paper.

K-means clustering (KM) is one of the simplest and most popular unsupervised machine learning algorithms [29] intending to group similar data points, and discover the underlying patterns. To achieve this goal, K-means looks for a fixed number (k) of clusters in a data set. The K-means algorithm identifies the k number of centroids, with k given in advance, and then assigns each data point to the nearest cluster, keeping the centroids as small as possible.

Hierarchical clustering (AC) is a method of cluster analysis which seeks to build a hierarchy of clusters. Hierarchical clustering can be created following a "bottom-up" approach, where each observation starts in its own cluster, and pairs of clusters are merged as one moves up the hierarchy. This strategy is called agglomerative. The opposite approach, divisive, starts with all observations in one cluster, and splits are performed recursively as one moves down the hierarchy. Merges and splits normally happen using a greedy algorithm, and the final result of the hierarchy of clusters can be visualized as a tree structure, called a dendrogram.

### 2.4. Keyphrases Identification

Clustering has grouped all the terms (actually their representation as word embedding vectors) into n clusters. Each cluster has a centroid which, only in the case of the AP algorithm, corresponds to an exemplar, i.e., an existing term. For the other algorithms, 1 or m terms closest to the centroid are extracted.

Then the candidate keywords, obtained in step 1, are compared with the m terms closest to the centroid, and the resulting terms are identified as key phrases. In the case of single words, only nouns are extracted; in the case of compound words, candidate keywords have the form of (1) or (2), respectively, for texts in English or Italian.

### 2.5. Evaluation

Once the keywords/keyphrases have been extracted, the task of evaluating the results is to be undertaken. In this case, we are interested in evaluating two different aspects. On the one hand, we are interested in comparing and evaluating the word embedding models, able to enhance the semantic aspects of keyword extraction. On the other hand, a comparison of this method with standard algorithms to extract keywords, such as RAKE, TextRank, TF-IDF on the English and Italian datasets, is shown. We use Sørensen–Dice (SD) (https://en.wikipedia.org/wiki/S%C3%B8rensen%E2%80%93Dice_coefficient) similarity coefficients, able to measure the shared information (overlap) over the sum of cardinalities. Its value is computed as follows:

$$SD = \frac{2|X \cap Y|}{|X| + |Y|} \tag{6}$$

where $|X|$ and $|Y|$ are the cardinality of the two sets, while $|X \cap Y|$ is the cardinality of the items that the two sets have in common.

## 3. Our Approach

### 3.1. Datasets

In choosing the datasets, we are interested in having Italian and English versions of data, with textual descriptions long enough to fit the purpose, and possibly tagged. Here a brief description of the four datasets used.

FilmTV movies dataset (https://www.kaggle.com/stefanoleone992/filmtv-movies-dataset), from now on referred to as Movie, provides the information on movies with data on how users from different countries rate the movies compared to each other. Data has been scraped from the publicly available website https://www.filmtv.it (accessed on 24 November 2019). Each row represents a movie available on FilmTV.it, with the original title, year, genre, duration, country, director, actors, average vote, and votes. Both versions (English and Italian) contain 53,497 movies, and the Italian version contains one extra-attribute for the local title used when the movie was published in Italy. The "description" of the movie has been used for extracting keywords. The length of the text is variable, ranging from a simple phrase, that gives some hints of the plot, to a more detailed explanation of the movie. The language, both in Italian and in English, is plain, without jargon or technicalities.

Food.com Recipes and Interactions dataset (https://www.kaggle.com/shuyangli94/food-com-recipes-and-user-interactions), from now on referred to as Recipe (English version), consists of 180K+ recipes and 700K+ recipe reviews covering 18 years of user interactions and uploads on Food.com (formerly GeniusKitchen). For our purpose, this dataset is perfect, having each recipe a textual description of the steps involved, and tags associated. The language used to outline the procedure is simple, but the nouns of kitchen ingredients and tools are rarely used in ordinary conversation.

Recipe (Italian version) is a dataset with 28,200 Italian recipes, with ingredients, and food preparation steps. The dataset was created by Giorgio Musilli (https://www.dbricette.it).

The preparation steps, used as input to extract keywords, use a basic language, with verbs to infinitive and some essential information on how to combine, cut, cook the ingredients.

The summary of the main information on the 4 datasets used, two in Italian and two in English, is shown in Table 2. The datasets are rather different, both in the length of the sentences and in the number of candidate words present.

**Table 2.** Information on the 4 datasets used.

|  | Recipe | Movie | Recipe | Movie |
| --- | --- | --- | --- | --- |
| Language | It | It | En | En |
| Document no. | 28,200 | 53,497 | 180,000 | 53,497 |
| Used | 555 | 555 | 555 | 400 |
| Av. sent. no. | 4.0 | 3.71 | 9.34 | 3.55 |
| Av. 1 g cand. words no. | 23.96 | 25.02 | 42.50 | 8.63 |
| Total 1 g cand. Words | 2336 | 13,889 | 23,588 | 3461 |
| Av. $n$-gram cand. words no. | 15.95 | 15.31 | 29.91 | 4.97 |
| Total $n$-gram cand. words | 8816 | 8516 | 14,378 | 1975 |

*3.2. Experiments*

The method depicted in Figure 1 has been implemented, according to the steps defined in Table 3, described in depth in their computational aspects.

**Table 3.** Steps of the proposed method.

#Preprocessing
- Preprocessing of n documents
- For each document in n documents:
    ○ Extract candidate keywords
#Semantic relatedness computation
- Extract a subset of word embedding limited to candidate keywords
- Compute semantic relatedness matrices
#Clustering
- Cluster candidate keywords (using Affinity Propagation, K-means, and hierarchical cluster)
- Extract k terms closest to the centroid for each cluster
#Keyword/keyphrases identification
- Identify as keywords the candidate keywords that match the k terms closest to the centroids.
#For baseline
- Preprocessing of n documents
- Compute TF-IDF on the n documents
- For each document in n documents
    ○ Compute RAKE and TextRank

The data have been pre-processed according to the standard pipeline, presented previously.

This preprocessing phase aims to extract from phases in plain language noun or noun phrases. The tool(s) identified and possibly integrated should, therefore, be able to understand the phrases structures and dependencies structure, to correctly identify noun phrases.

Tools present in the literature have been identified and tested: for the English language, datasets were processed using standard tools, such as Stanford's core NLP suite, Natural Language Toolkit NLTK [30] of Python, with PENN Treebank [31] as a POS tagger and tokenizer.

Stanford's core NLP suite offers a set of tools for managing natural Languages. It can give the base forms of words, their parts of speech (POS), normalize dates, times, and numeric quantities, and identify noun phrases and mark up the structure of sentences in terms of phrases and syntactic dependencies [32].

Natural Language Toolkit NLTK is a tool able to work with human language data. It provides a suite of text processing libraries for classification, tokenization, stemming, tagging, parsing, and semantic reasoning.

Spacy (https://spacy.io/) and Pattern (https://www.uantwerpen.be/en/research-groups/clips/), two further python packages for NLP, have been tested too.

SpaCy is an open-source software library for advanced NLP and features convolutional neural network models for part-of-speech tagging, dependency parsing, and named entity recognition (NER). It offers pre-built statistical neural network models (also for the Italian language) as well as tools to train custom models on their own datasets.

Pattern, developed by Clips (Computational Linguistics & Psycholinguistics) at the University of Antwerp, is a Python package for web mining, natural language processing, machine learning, and network analysis, with a focus on ease-of-use [33].

For the preprocessing of the Italian datasets, some preliminary tests have been performed with the NLTK package, the Pattern python package specific for Italian, and Spacy (with pre-trained Italian model), but they do not provide acceptable results. Stanford core NLP has not developed tools for the Italian language.

The main problems are encountered with POS tagging, probably due to an incorrect syntactic interpretation of the sentence: for example, adjectives, which have the same form of a verb, are misinterpreted, and therefore possible noun phrases are not identified.

The tool that proved to work best is the TreeTagger [21], a free tool developed by Helmut Schmid at the Institute for Computational Linguistics of the University of Stuttgart, using the standard Italian tagset, although it still has some issues for both POS tagging and lemmatization.

To identify *n*-gram words, different procedures and tools were tested. In the end, based on experimental evaluations, we started from a grammatical analysis by TreeTagger, able to identify grammatical structures in sentences and dependencies, and integrated the results with NLTK and dependency parsing tools: in this way, integrating NLTK and TreeTagger, it was possible to identify *n*-grams words that appear in documents.

After POS tagging, single words that are nouns and compound words that are nouns and adjectives (according to the grammatical rules of each language) were held on the four datasets. Table 4 reports the total number of candidate keywords and the number of different words, either single or compound.

**Table 4.** Dimensions of pre-trained models used.

| Model | Italian | English |
|---|---|---|
| W2v wiki | 1,721,340 | 4,530,030 |
| W2v google | 50,031 | 3,000,000 |
| GloVe | 730,613 | 400,000 |

To estimate the semantic relatedness of words and to compute their similarity, we have used Word2Vec and GloVe pre-trained models in English and Italian. In detail the models used:

- Word2vec trained on google news:

  ○ the English version includes word vectors for a vocabulary of 3 million words and phrases that have formed on about 100 billion words of a Google News dataset (GoogleNews-vectors-negative300.bin.gz).

  ○ the Italian version has been trained on Italian Wikipedia by ISTI-CNR, trained with skipgram (http://hlt.isti.cnr.it/wordembeddings/)

- Word2vec trained on Wikipedia (https://wikipedia2vec.github.io/wikipedia2vec/pretrained/) both Italian and English versions have been trained on a Wikipedia version of April 2018, with the following parameters:

- ○    –dim-size = 300: The number of dimensions of the vectors.
- ○    –window = 5: The maximum distance between the target item (word or entity) and the context word to be predicted.
- ○    –iteration = 10: The number of iterations for Wikipedia pages.
- ○    –negative = 15: The number of negative samples.

- GloVe pre-trained vectors:

  - ○    the English version has been trained on Wikipedia 2014 + Gigaword 5, for a total of 6 Billion tokens, 400 K vocabulary (https://nlp.stanford.edu/projects/glove)
  - ○    the Italian version has been trained on Italian Wikipedia by ISTI–CNR.

In Table 4, the dimension of pre-trained word embedding models used is shown.

For all models used, the vector length is 300 features.

From the pre-trained models, the vectors corresponding to the candidate keywords have been extracted. The vectors of the *n*-gram words have been calculated as an average of the vectors of the single words, if present in the model, according to the formula defined above in Equation (3).

In Appendix B, Table A5 details the percentage of coverage, that is how many candidate keywords are present in the word embedding model considered, how many terms are present in the model (in model), and how many are out of vocabulary (oo vocab) terms. Table A5 reports data for 1-g and *n*-grams, respectively.

The out of vocabulary terms, that terms not present in the models, can be grouped in three main categories:

1. Terms with some typos, as "washignton" or "agosciosa", the n of "angosciosa" is missing (anguished) or "areoporto" misspelled for airport. This category of terms can be automatically corrected using a tool able to measure the distance of two words, e.g., based on character exchange.
2. Technicalities or sector-specific terms, such as "chorizos" from recipe/en or "brunoise" (of French origin, indicating a technique for cutting vegetables) from recipe/it. A training phase to add these terms to the word embedding models should be undergone to deal with these cases.
3. Diminutives or nicknames, which are very frequent in Italian, as "padellino" (that mean little saucepan) or "ziette", aunts or "storiellina", nice little stories. A lemmatization/stemmer tool should be trained to be able to deal with these kinds of modification of words.

It is worth noting that in general w2v wiki and GloVe have greater coverage, while w2v google has many terms outside the vocabulary, due probably to the low number of terms in the models. This is especially valid for the pre-trained model on google news in Italian. The movie datasets, both in Italian and English, show a very high percentage of coverage, because of the common words used in the description, while the recipes, to different extents, suffer the problem of the kitchen-specific words. The recipe/en generally exceeds for out-of-vocabulary terms and is the only case where the number of oo terms is higher for the w2vwiki model than for GloVe.

Then, semantic relatedness matrices were calculated, based respectively on the measurement of cosine similarity and Euclidean distance, able to measure the relatedness of any two words, based on embedded vectors.

The cluster algorithms have been applied to the similarity matrices, related to the terms (and their vectors) of the candidate keywords, using different criteria of similarity, cosine similarity, and Euclidean distance. The algorithms used were: Affinity Propagation, K-means, and hierarchical clustering. K-means and hierarchical clustering algorithms require the number of clusters as input: therefore, the number of clusters obtained by the Affinity Propagation on cosine similarity measurements and Euclidean distance has been inserted in these cases.

Table 5 reports results on the clustering process, using the clustering algorithms on the four datasets, with the word embedding models.

**Table 5.** Number of clusters per models and per similarity metrics found by the Affinity Propagation (AP) algorithm, for 1-g and *n*-grams.

| Models | 1-g | Recipe/It | Movie/It | Recipe/En | Movie/En |
|---|---|---|---|---|---|
| W2v wiki | Vocabulary | 1763 | 5575 | 2101 | 1812 |
| | AP Cosine similarity | 154 | 541 | 235 | 221 |
| | AP Euclidean distance | 42 | 171 | 62 | 121 |
| W2v google | Vocabulary | 1219 | 4636 | 2076 | 1678 |
| | AP Cosine similarity | 121 | 386 | 265 | 219 |
| | AP Euclidean distance | 107 | 300 | 85 | 96 |
| GloVe | Vocabulary | 1740 | 5529 | 2184 | 1792 |
| | AP Cosine similarity | 190 | 550 | 229 | 193 |
| | AP Euclidean distance | 190 | 550 | 169 | 112 |
| **Model** | **n-grams** | **Recipe/it** | **Movie/it** | **Recipe/en** | **Movie/en** |
| W2v wiki | Vocabulary | 3448 | 8595 | 6164 | 2594 |
| | AP Cosine similarity | 461 | 1364 | 795 | 485 |
| | AP Euclidean distance | 453 | 915 | 791 | 301 |
| W2v google | Vocabulary | 2917 | 7734 | 6142 | 2454 |
| | AP Cosine similarity | 367 | 923 | 735 | 473 |
| | AP Euclidean distance | 436 | 891 | 734 | 303 |
| GloVe | Vocabulary | 3426 | 8551 | 6254 | 2574 |
| | AP Cosine similarity | 495 | 1320 | 666 | 436 |
| | AP Euclidean distance | 343 | 889 | 680 | 301 |

Our method, starting from candidate keywords, identifies those words that are centroid or n terms (1-g or *n*-gram) closer to the centroid. If the number of clusters is higher, the extracted keywords are more numerous.

In general, it can be noted that the number of clusters obtained depends greatly from the word embedding model used, hence on the number of terms usable, and is higher for the Cosine similarity matrix with respect to Euclidian distance.

The last step is the extraction of keyphrases to be associated with each text. We have matched candidate keywords/keyphrases extracted from the text, eventually in the form of (1) or (2), with the term(s) closest to the centroid of the clustering algorithms. In particular, for Affinity Propagation, the centroid is exactly the exemplar, while for K-mean and hierarchical clustering, the k terms closest to the centroid (calculated, if not available, as the average of its members) have been identified. For greater homogeneity and to allow a better comparison among the algorithms, also for Affinity Propagation have been extracted, besides the exemplar, the k-1 closest terms.

## 4. Discussion

The evaluation metrics for automatic unsupervised keyphrase extraction methods is a complex task. In general, the results are compared with the keywords chosen manually by experts or using gold standard datasets. In [20], we used Recall and Precision to evaluate the results, the same metrics used in SemEval contexts (https://en.wikipedia.org/wiki/SemEval). Here our idea is to define a fully automatic evaluation method, to analyze the different options tested in the runs, and compare and evaluate the results.

We are interested in evaluating how much the word embedding models and the similarity measures influence the results. We would also like to measure how our approach that combines word embedding with cluster algorithms obtains comparable results with methods such as TF-IDF, TextRank, and RAKE. For this purpose, we used Sørensen–Dice (SD) similarity coefficient computed as described above in Equation (6).

Figure 3 shows the SD similarity coefficient for the word embedding models for Affinity Propagation, K-means, and hierarchical clustering, using Cosine similarity matrix, on 1-g, on the left,

and *n*-grams, on the right. Due to the higher number of clusters, and hence, of keywords/keyphrases extracted for *n*-grams, the SD similarity coefficient is higher for *n*-grams than for single words.

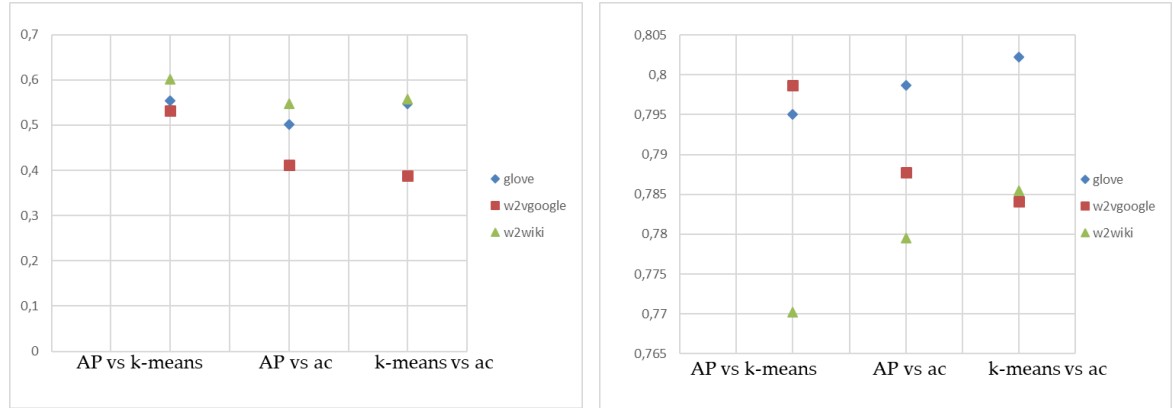

**Figure 3.** Comparison of clustering algorithms on 1-g (on the **left**) and *n*-grams (on the **right**).

Figure 4 compares the same clustering algorithm with the two similarity measures used. In the case of single words, the similarity coefficient Sørensen–Dice (SD) is very low, due to the limited number of extracted keywords, while the coefficient for compound words ranges from 0.4 to 0.7. This means that clustering algorithms always extract at least one or probably more common keyphrases from the texts.

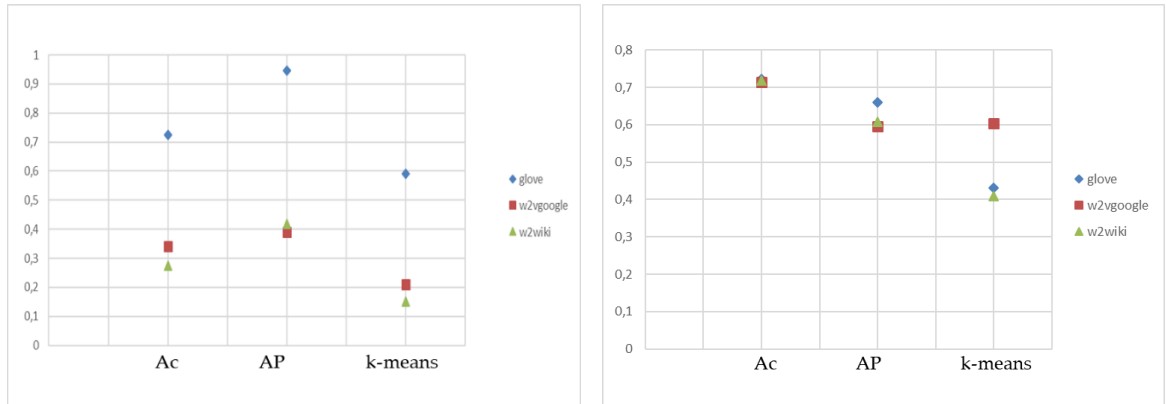

**Figure 4.** Comparison of similarity measures (Cosine similarity vs. Euclidean distance) on 1-g (on the **left**) and *n*-grams (on the **right**).

Figures 5–7 compare the results obtained by the proposed method with the standard approaches, which are Rake, TextRank, and TF-IDF. The graphs show that neither cluster algorithms nor word embedding models produce significant variations in the results: the values of the SD coefficient have almost stacked. The choice and the cardinality of keyword sets against which to evaluate the results greatly influence the evaluation: in these tests, we used all those extracted from the different algorithms. The results show that for TextRank, RAKE, and TF-IDF, there is almost always some overlap.

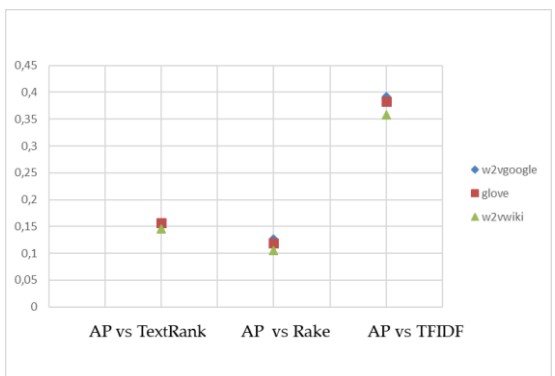
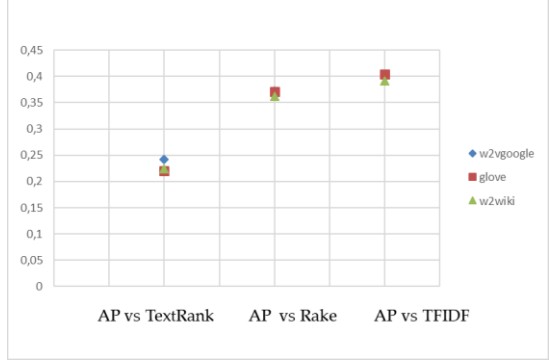

**Figure 5.** Comparison of results of this method (using AP clustering and Cosine similarity) with baseline on 1-g (on the **left**) and *n*-grams (on the **right**).

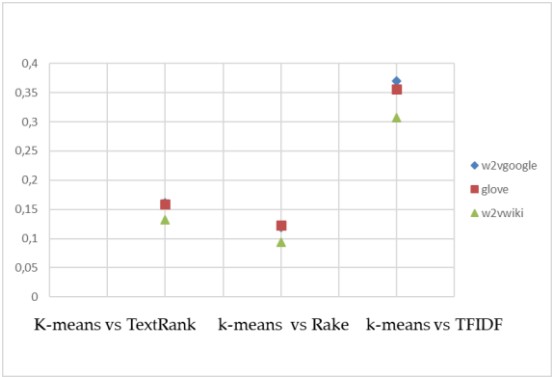
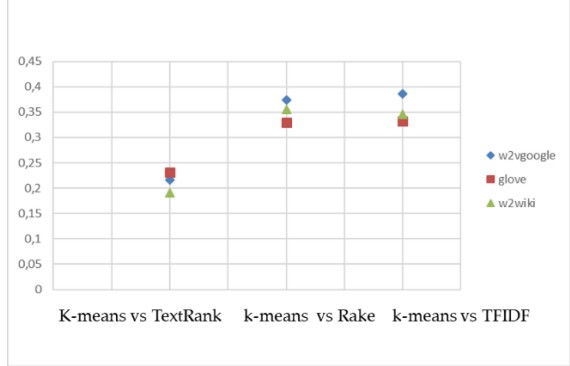

**Figure 6.** Comparison of results of this method (using k-means clustering and Cosine similarity) with baseline on 1-g (on the **left**) and *n*-grams (on the **right**).

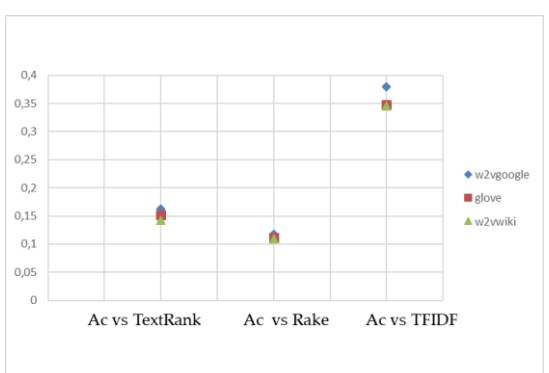
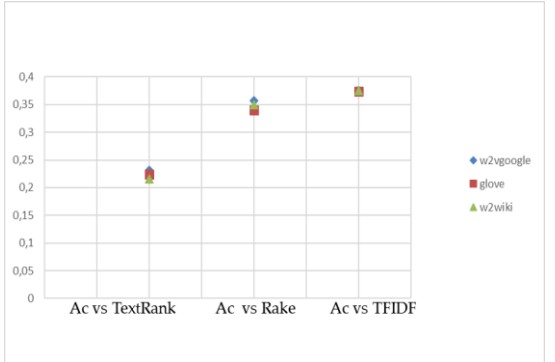

**Figure 7.** Comparison of results of this method (using hierarchical clustering and Cosine similarity) with baseline on 1-g (on the **left**) and *n*-grams (on the **right**).

Analyzing the extracted terms, it can be noticed that the method extracts significant terms, able to describe the content of the texts, but, especially in the case of recipes in the Italian language, since many culinary terms are grouped in the same cluster, the ability to describe different aspects is lost. When using the 4 terms, this problem is partially overcome.

Movie/it dataset has been used for the graphs of Figures 3 and 4, and Recipe/en dataset for the graphs of Figures 5–7. The other datasets provide similar results, not reported here. Appendix A reports the data used to create the graphs.

For the proposed method, the time complexity of the different experiments depends on the complexity of the cluster algorithms used (see Table 3), with the same dataset and word embedding model. The experiments have been performed on a DELL PC, with Intel Core i7-8565u 1.8 GHz

1.99 GHz with a memory of 16GB, using Windows 10 operating system. In Table 6, we report the computational costs for keyword/keyphrases extraction method performed on 100 documents of (Italian and English) recipe datasets and 500 documents of (Italian and English) movie datasets. All steps line presents the computation costs for the complete execution of the proposed approach, including the baseline algorithms. The matrix + clustering + keyword extraction row reports the computational time for the method proposed, using all the clustering algorithms (Affinity Propagation, K-means, and hierarchical clustering), applied to the adopted semantic similarity matrices (Euclidean distance and Cosine similarity), specifying the word embedding model and the granularity (1-g or $n$-grams). In the following lines are reported the specific results for the w2vecwiki, glove, and w2vgoogle word embedding models.

**Table 6.** Computational costs for the proposed method.

| Computational Costs (In Seconds) | Recipe 100 Docs | | | Movie 500 Docs | | |
|---|---|---|---|---|---|---|
| | Min | Max | Mean | Min | Max | Mean |
| all steps | 243.90 | 1935.06 | 828.70 | 1051.59 | 13,218.80 | 4437.71 |
| baseline | 24.19 | 41.45 | 31.92 | 283.08 | 627.23 | 428.03 |
| preprocessing | 0.15 | 0.69 | 0.31 | 0.36 | 1.21 | 0.62 |
| candidates key | 0.36 | 140.36 | 68.61 | 3.22 | 666.17 | 333.47 |
| extract subset of w_e | 4.17 | 1.453.88 | 479.19 | 3.96 | 3222.26 | 777.72 |
| matrix + clustering + keyword extraction | 79.89 | 456.16 | 248.36 | 92.99 | 9774.32 | 2894.30 |
| matrix + clustering + keyword extraction: using w2vwiki | 107.66 | 454.95 | 275.86 | 171.82 | 9472.07 | 2942.04 |
| matrix + clustering + keyword extraction: using glove | 99.74 | 408.33 | 257.99 | 257.84 | 9026.27 | 2872.33 |
| matrix + clustering + keyword extraction: using glove w2vgoogle | 78.93 | 365.40 | 207.98 | 89.92 | 9763.60 | 9763.60 |

The experimental setup has been implemented in Python 3.7, using standard packages like Numpy, Matplotlib, Pandas, and other more specific ones for processing of textual data such as NLTK, Treetagger, Gensim [34], Newspaper, Pattern (Pattern clips 2.6), and Sklearn, together with some experimental packages in GitHub. We use various software packages to implement TextRank, RAKE, and Word2Vec [34] using standard options.

## 5. Conclusions

This paper presents an unsupervised automatic keyword extraction method based on clustering integrating different word embedding models to improve the semantic relatedness of keywords. The main characteristics of this approach are the integration of word embedding models, able to capture the semantics of words and their context, with the clustering algorithms, able to identify the essence of the terms. The more significant one(s), able to represent the contents of a text, are then chosen. The whole process, the possible choices, and the method implemented, together with datasets, and some preliminary results obtained have been described. The results presented have been obtained with pre-trained word embedding models on both Italian and English datasets.

Mainly for Italian, especially in the case of such specific topics as recipes, the pre-trained models are struggling to be adequate and to bring correct results. A further source of errors is POS tagging, always for the Italian language: although the solution adopted works quite well, there are still errors, especially in the dataset of the recipes, due to the incorrect identification of the grammatical structure of the phrases.

Aspects that will be furtherly studied and analyzed in the future concern:

- Test and integrate other pre-processing tools, specifically designed for the Italian language for better POS tagging and lemmatization results.
- Define and apply tools capable of handling typographical errors, and out-of-vocabulary words, to improve coverage and increase candidate keywords.

- Test our approach on texts in French, German, evaluating the results obtained, and the effort needed to adapt it.
- Test FastText and other word embedding models
- Adapt and custom pre-trained embedding models with the inclusion of new terms, to add out-of-vocabulary words and update weights, with the aim to address the issue of specialist words or contexts.
- Here the cluster algorithms have been used individually, one at a time: it is planned to test and evaluate the possibility of integrating clustering methods to adjust accordingly keyphrases weights seamlessly: the weight of the common terms in the clustered documents is increased, while the others are decreased, also based on their representativeness in other clusters.
- Improve/integrate evaluation methods, able to take into account the different aspects of the problem.
- Integrate Wikipedia, the largest encyclopedia collected and organized by the human on the web, as the knowledge base to measure term relatedness.

**Author Contributions:** Methodology, I.G.; software, I.G. and M.T.A.; data curation, I.G. and M.T.A.; writing—original draft preparation, I.G.; writing—review and editing, M.T.A. All authors have read and agreed to the published version of the manuscript."

**Funding:** This research received no external funding.

**Conflicts of Interest:** The authors declare no conflict of interest.

## Appendix A

Tables A1 and A2 report SD coefficients for the comparison among word embedding models and clustering algorithms. The values in the tables have been used for the graphs reported in Figures 3 and 4.

**Table A1.** Sørensen–Dice (SD) coefficients for word embedding models and clustering algorithms, on 1 g.

| Cosine Similarity | AP vs. *k*-means | Ap vs. Hierarchical | *k*-means vs. Hierarchical |
|---|---|---|---|
| **glove** | 0.553753 | 0.501133426 | 0.546715 |
| **w2vgoogle** | 0.531015 | 0.411937812 | 0.388053 |
| **w2wiki** | 0.60165 | 0.547461553 | 0.557486 |
| **Euclidean distance** | **AP vs. k-means** | **Ap vs. hierarchical** | **k-means vs. hierarchical** |
| **glove** | 0.372404 | 0.510199731 | 0.590246 |
| **w2vgoogle** | 0.490649 | 0.554472764 | 0.209238 |
| **w2wiki** | 0,342949 | 0.486245837 | 0.150821 |
| | **AP cosine vs. *k* Euclidean** | **AP cosine vs. ac Euclidean** | ***k* cosine vs. ac Euclidean** |
| **glove** | 0.401407 | 0.675455028 | 0.70793 |
| **w2vgoogle** | 0.60148 | 0.713807828 | 0.687272 |
| **w2wiki** | 0.376982 | 0.698720215 | 0.684565 |
| | **ac cosine vs. ac Euclidean** | **AP cosine vs. AP Euclidean** | ***k*-means cosine vs. *k*-means Euclidean** |
| **glove** | 0.725692 | 0.94656135 | 0.590246 |
| **w2vgoogle** | 0.341519 | 0.389750186 | 0.209238 |
| **w2wiki** | 0.275528 | 0.419050416 | 0.150821 |
| | **AP Euclidean vs. *k*-means cosine** | **AP Euclidean vs. ac cosine** | ***k* Euclidean vs. ac cosine** |
| **glove** | 0.601624 | 0.582922191 | 0.422509 |
| **w2vgoogle** | 0.530952 | 0.551179024 | 0.58458 |
| **w2wiki** | 0.545754 | 0.521521319 | 0.382657 |

**Table A2.** SD coefficients for word embedding models and clustering algorithms, on *n* gram.

| Cosine Similarity | AP vs. *k*-Means | Ap vs. Hierarchical | *k*-Means vs. Hierarchical |
|---|---|---|---|
| glove | 0.795003838 | 0.798703 | 0.802229 |
| w2vgoogle | 0.798716333 | 0.787743 | 0.784118 |
| w2wiki | 0.770255233 | 0.779518 | 0.785443 |
| **Euclidean distance** | **AP vs. *k*-means** | **Ap vs. hierarchical** | ***k*-means vs. hierarchical** |
| glove | 0.372404424 | 0.5102 | 0.451053 |
| w2vgoogle | 0.490648924 | 0.554473 | 0.641539 |
| w2wiki | 0.342949357 | 0.486246 | 0.419902 |
| | **AP cosine vs. *k* Euclidean** | **AP cosine vs. ac Euclidean** | ***k* cosine vs. ac Euclidean** |
| glove | 0.401407 | 0.675455028 | 0.70793 |
| w2vgoogle | 0.60148 | 0.713807828 | 0.687272 |
| w2wiki | 0.376982 | 0.698720215 | 0.684565 |
| | **ac cosine vs. ac Euclidean** | **AP cosine vs. AP Euclidean** | ***k*-means cosine vs. *k*-means Euclidean** |
| glove | 0.659123 | 0.401407 | 0.675455 |
| w2vgoogle | 0.596285 | 0.60148 | 0.713808 |
| w2wiki | 0.608533 | 0.376982 | 0.69872 |
| | **AP Euclidean vs. *k*-means cosine** | **AP Euclidean vs. ac cosine** | ***k* Euclidean vs. ac cosine** |
| glove | 0.601624 | 0.582922 | 0.422509 |
| w2vgoogle | 0.530952 | 0.551179 | 0.58458 |
| w2wiki | 0.545754 | 0.521521 | 0.382657 |

Tables A3 and A4 report SD coefficients for the comparison between the baseline approaches and our algorithm evaluated on the basis of the specific clustering algorithm. The values in the tables have been used for the graphs reported in Figures 5–7.

**Table A3.** SD coefficients for baseline approaches and our method, on 1 g.

| Cosine | AP vs. TextRank | AP vs. Rake | AP vs. TF-IDF |
|---|---|---|---|
| w2vgoogle | 0.154031 | 0.12685 | 0.392171 |
| glove | 0.157153 | 0.119307 | 0.382592 |
| w2vwiki | 0.145897 | 0.105691 | 0.357797 |
| **cosine** | ***k*-means vs. TextRank** | ***k*-means vs. rake** | ***k*-means vs. TF-IDF** |
| w2vgoogle | 0.160908 | 0.119247 | 0.370346 |
| glove | 0.158661 | 0.122957 | 0.356506 |
| w2vwiki | 0.132477 | 0.093128 | 0.307014 |
| **cosine** | **Ac vs. TextRank** | **Ac vs. rake** | **Ac vs. TF-IDF** |
| w2vgoogle | 0.161919 | 0.117956 | 0.379477 |
| glove | 0.151876 | 0.112269 | 0.3481 |
| w2vwiki | 0.142518 | 0.108938 | 0.345622 |
| **Euclidean** | **AP vs. TextRank** | **AP vs. rake** | **AP vs. TF-IDF** |
| w2vgoogle | 0.108412 | 0.054019 | 0.188562 |
| glove | 0.146281 | 0.102226 | 0.309676 |
| w2vwiki | 0.108963 | 0.041356 | 0.172741 |
| **Euclidean** | ***k*-means vs. TextRank** | ***k*-means vs. rake** | ***k*-means vs. TF-IDF** |
| w2vgoogle | 0.00317 | 0.012068 | 0.046226 |
| glove | 0.1148 | 0.11094 | 0.25953 |
| w2vwiki | 0.004497 | 0.006122 | 0.019548 |
| **Euclidean** | **Ac vs. TextRank** | **Ac vs. rake** | **Ac vs. TF-IDF** |
| w2vgoogle | 0.049533 | 0.045547 | 0.178766 |
| glove | 0.105825 | 0.087086 | 0.28053 |
| w2vwiki | 0.02561 | 0.030079 | 0.144307 |

**Table A4.** SD coefficients for baseline approaches and our method, on *n* gram.

| Cosine | AP vs. TextRank | AP vs. Rake | AP vs. TF-IDF |
|---|---|---|---|
| **w2vgoogle** | 0.241973 | 0.372759 | 0.401519 |
| **glove** | 0.220545 | 0.371246 | 0.403963 |
| **w2vwiki** | 0.224445 | 0.362539 | 0.390922 |
| **cosine** | ***k*-means vs. TextRank** | ***k*-means vs. rake** | ***k*-means vs. TF-IDF** |
| **w2vgoogle** | 0.216192 | 0.374533 | 0.386397 |
| **glove** | 0.230609 | 0.32928 | 0.332172 |
| **w2vwiki** | 0.191115 | 0.355518 | 0.345643 |
| **cosine** | **Ac vs. TextRank** | **Ac vs. rake** | **Ac vs. TF-IDF** |
| **w2vgoogle** | 0.231572 | 0.357224 | 0.375333 |
| **glove** | 0.22371 | 0.340309 | 0.373955 |
| **w2vwiki** | 0.216129 | 0.350167 | 0.374979 |
| **Euclidean** | **AP vs. TextRank** | **AP vs. rake** | **AP vs. TF-IDF** |
| **w2vgoogle** | 0.177379 | 0.299848 | 0.313574 |
| **glove** | 0.181597 | 0.281195 | 0.299125 |
| **w2vwiki** | 0.191386 | 0.304227 | 0.33789 |
| **Euclidean** | ***k*-means vs. TextRank** | ***k*-means vs. rake** | ***k*-means vs. TF-IDF** |
| **w2vgoogle** | 0.214267 | 0.31975 | 0.316057 |
| **glove** | 0.186545 | 0.281751 | 0.272817 |
| **w2vwiki** | 0.207366 | 0.322137 | 0.321478 |
| **Euclidean** | **Ac vs. TextRank** | **Ac vs. rake** | **Ac vs. TF-IDF** |
| **w2vgoogle** | 0.197352 | 0.353842 | 0.378546 |
| **glove** | 0.225183 | 0.330543 | 0.360484 |
| **w2vwiki** | 0.21442 | 0.365657 | 0.38084 |

## Appendix B

In Table A5 percentage of coverage, that is how many candidate keywords are present in the word embedding model considered, how many terms are present in the model (in model), and how many are out of vocabulary (oo vocab) terms. Table A5 report data for 1-g and *n*-grams, respectively.

**Table A5.** Statistics for embedding models on datasets, using 1-g and *n*-grams.

| 1-g | Recipe/It | Movie/It | Recipe/En | Movie/En |
|---|---|---|---|---|
| Cand. keywords | 1819 | 5663 | 2394 | 1826 |
| W2v wiki coverage | 96.92% | 98.45% | 87.76% | 99.23% |
| W2v wiki in-model | 1763 | 5575 | 2101 | 1812 |
| W2v wiki oo vocab | 56 | 88 | 293 | 14 |
| W2v google coverage | 67.01% | 81.86% | 86.72% | 91.89% |
| W2v google in-model | 1219 | 4636 | 2076 | 1826 |
| W2v google oo vocab | 600 | 1027 | 318 | 148 |
| GloVe coverage | 95.66% | 97.63% | 91.23% | 98.14% |
| GloVe in-model | 1740 | 5529 | 2184 | 1792 |
| Glove oo vocab | 79 | 134 | 210 | 34 |
| ***n*-grams** | **Recipe/it** | **Movie/it** | **Recipe/en** | **Movie/en** |
| W2v wiki coverage | 98.60% | 99.16% | 95.74% | 99.46% |
| W2v wiki in-model | 3448 | 8595 | 6164 | 2594 |
| W2v wiki oo vocab | 49 | 73 | 271 | 14 |
| W2v google coverage | 83.41% | 89.22% | 95.40% | 94.10% |
| W2v google in-model | 2971 | 7734 | 6142 | 2454 |
| W2v google oo vocab | 561 | 909 | 293 | 146 |
| GloVe coverage | 97.97% | 98.65% | 97.14% | 98.03% |
| GloVe in-model | 3426 | 8551 | 6254 | 2574 |
| Glove oo vocab | 71 | 117 | 184 | 34 |

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
