# Peer review of "Semantic Unsupervised Automatic Keyphrases Extraction by Integrating Word Embedding with Clustering Methods"

_mti, doi:10.3390/mti4020030_

Round 1
Reviewer 1 Report
Authors claim that the paper presents a novel algorithm to address the problem of the automatic unsupervised extraction of keywords/phrases from texts. They use well know vector representations for words, plus a clustering algorithm to group similar keywords and choose keywords, or key phrases, that represent the most significant clusters of the document content.
More than an algorithm the paper presents a method that uses word embedding and clustering to extract document keywords or key phrases. In fact, many other authors use similar approaches, resulting in a very reduced novelty of the work.
In strict terms of the paper scope and objectives, although the problem is well defined, authors lack to point exactly the strengths of their method against others available in the literature. Results and discussion are focused on benchmarking the combinations of w2v/glove with clustering (k-means, AP, HC). Paper also lacks the performance analysis in terms of time complexity for the different evaluated setups.
Several other paper issues shall also be corrected:
- Many bibliographic references missing
- Structure of the paper shall be improved, many sub-sections, many tables, many figures.
- Balancing between section size shall also be revised. A section about, methods and techniques or algorithm shall be enhanced. Little or no theoretical background is given, just a block diagram.
- English shall be revised.
Reviewer 2 Report
This is a very interesting and timely paper as we seek to find a lightweight approach to mine keywords, rather than use cumbersome ontologies.
I would like the authors to back up their claims e.g. The tools identified for the Italian language allow us to easily adapt them to other 67 languages such as French, German, etc...
It might be easy to process different languages, but due to their structure will it produce similar results to English an Italian?
I have questions over how you can validate the results as you are only using a proxy from Wikipedia data. What is the reliability of using this method to determine the accuracy of the results compared to human determined Golden Data?
I would like to see a greater explanation of the datasets chosen as given they are quite niche I would imagine the results would be very favourable. Is this a fair comparison against the other techniques and the dataset they used / should be used on? However, your research does show it works for movies and food (noting the golden data validity above).
It is interesting to note in the results it doesn’t deal with vernacular slang well, or you deem it not to deal with it well even though it highlights it. Clearly there are misspellings which it picks up.
Overall it’s an interesting paper, but the results require further validation / greater explanation as discussed above.
Comments attached to PDF as well.

Reviewer 3 Report
The paper presents a technique for unsupervised extraction of keywords expressed in both English and Italian using a combination of Word2Vec and GloVe techniques. The problem of extracting keywords across multiple languages is important and this paper attempts to address the part.
The advantage is that these are easy to replicate to a great extent. The downside is that these are both well known techniques, so the novelty factor is rather low.
Many references/internal references are marked as `error! reference source not found'! Was this paper proof read before being submitted? This makes the paper very hard to read.
The introduction requires a thorough rewrite. There is no information about why this method solves the problems that existed in prior approaches, or even what the problems of the prior art were. As such, this technique needs proper motivation.
Please describe the pre-processing phase in more detail. As it is, it just throws out a bunch of references that are hardly properly motivated.
Some of the results shown here seem interesting, but the paper requires a proper rewrite. Please focus on giving he exact steps you followed. As it stands, the paper is incomprehensible.
Round 2
Reviewer 3 Report
The authors have added in the clarifications sought and made the paper much nicer to read. The experiments are interesting. However, the overall novelty of the algorithm is not very high.
Author Response
We are thankful to the editor, the Academic editor, and anonymous reviewers for reviewing our article and for the valuable comments and suggestions.
We thank the reviewer for acknowledging the effort we put on revising and improving the paper.